**Data Availability Statement:** All relevant data are within the paper.

**Funding:** This work is supported by the National Natural Science Foundation of China under grant no. 31660678.

# Image classification of forage grasses on Etuoke Banner using edge autoencoder network

Ding Han[1,2], Minghua Tian[1], Caili Gong[1], Shilong Zhang[1], Yushuang Ji[1], Xinyu Du[1], Yongfeng Wei [1]*, Liang Chen[1]

**1** Information and Communication Engineering, Inner Mongolia University, Inner Mongolia Autonomous Region, China, **2** State Key Laboratory of Grassland Livestock Reproduction Regulation and Breeding, Inner Mongolia Autonomous Region, China

* weiyongfeng@imu.edu.cn

## Abstract

Automatically identifying the forage is the basis of intelligent fine breeding of cattle and sheep. In specific, it is a key step to study the relationship between the type and quantity of forage collected by cattle and sheep and their own growth, cashmere fineness, milk quality, meat quality and flavor, and so on. However, traditional method mainly rely on manual observation, which is time-consuming, laborious and inaccurate, and affects the normal grazing behavior of livestock. In this paper, the optimized Convolution Neural Network (CNN): edge autoencoder network(E-A-Net) algorithm is proposed to accurately identify the forage species, which provides the basis for ecological workers to carry out grassland evaluation, grassland management and precision feeding. We constructed the first forage grass dataset about Etuoke Banner. This dataset contains 3889 images in 22 categories. In the data preprocessing stage, the random cutout data enhancement is adopted to balance the original data, and the background is removed by employing threshold value-based image segmentation operation, in which the accuracy of herbage recognition in complex background is significantly improved. Moreover, in order to avoid the phenomenon of richer edge information disappearing in the process of multiple convolutions, a Sobel operator is utilized in this E-A-Net to extract the edge information of forage grasses. Information is integrated with the features extracted from the backbone network in multi-scale. Additionally, to avoid the localization of the whole information during the convolution process or alleviate the problem of the whole information disappearance, the pre-training autoencoder network is added to form a hard attention mechanism, which fuses the abstracted overall features of forage grasses with the features extracted from the backbone CNN. Compared with the basic CNN, E-A-Net alleviates the problem of edge information disappearing and overall feature disappearing with the deepening of network depth. Numerical simulations show that, compared with the benchmark VGG16, ResNet50 and EfficientNetB0, the $f1-score$ of the proposed method is improved by 1.6%, 2.8% and 3.7% respectively.

**Competing interests:** The authors have declared that no competing interests exist.

## Introduction

The traditional monitoring method of grazing behavior of cattle and sheep mainly relies on continuous and long-term artificial observation records. The data obtained is classified [1] and counted manually to measure animal intake. This approaches are strongly subjective and high labor intensive. Accurate and efficient identification of forage species is an important technology to study the relationship among the species, quantity and growth of plants [2], feed types and ecological balance [3] in the grazing behavior of cattle and sheep [3].

Data preprocessing steps are generally required before training a CNN model. Commonly, the methods of data preprocessing include: color space conversion, standardization, scale transformation, dimensionality reduction, image segmentation [4–6] and so on. Converting color space from RGB to HSV is an effective way to extract green features of grass, which ignores the natural disturbances and increases the robustness to light in image segmentation. Additionally, due to the number of species of each forage grasses is not uniform, most researchers use the balanced dataset or the cutout [7] method to process the unbalanced dataset [8, 9]. Therefore, the model improves the generalization ability and is not limited to some characteristics [10].

Apart from the above data preprocessing methods, biomorphology plays an important role in the identification of plant species, especially the edge and texture characteristics of grass plants. In the past, researchers used gradient [11], line detection, Laplacian, Sobel [12], gabor [13], Curvelet [14] to extract edges or texture features of grasses, the extracted features are identified by machine learning algorithms. Forage has not only edge features, but also rich texture features. Only depending on the edge, texture or color features of forage for classification, the high-level semantic information will not be effectively extracted by machine learning algorithms although each forage has unique edge texture features. This will greatly reduce the generalization ability of network model.

Attention mechanism has made ground-breaking achievements in the field of image classification. By extracting inter-channel, inter-channel and space, long-range, it can improve the convergence speed of network model, reduce network model parameters and increase network accuracy [15–18]. These attention mechanisms belong to soft learning mechanisms, which makes the adaptive learning of network model a significant feature for the final classification effect. The autoencoder network can be used as a tool to extract high-dimensional features [19], and it also has the function of attention mechanism to obtain more information about the main characteristics of grasses by CNN [20–22]. In this paper, the modified U-Net [23] is adopted as the autoencoder network. By fusing the abstract global information of grasses with the features of the backbone feature extraction network, a hard attention mechanism for the overall features of grasses is added.

In order to make a better use of biomorphic information of grazing images, hard attention mechanism is used to improve classification efficiency. In addition, a Sobel operator is employed to extract edge features of grazing images and a pre-training autoencoder network serves to extract abstract global features. Subsequently, the obtained grazing features are combined with the basic CNN algorithm, namely, edge autoencoder network(E-A-Net). The algorithm utilizes multi-scale edge features [24] to prevent the low-level characteristics of forage grasses edge from disappearing. The autoencoder network effectively extracts the abstract global features to prevent the CNN from focusing only on the significant features of forage grasses and ignoring the overall features.

There are three main innovations in this paper 1: This is the first time to study the forage grass dataset about Etuoke Banner. 2: We propose an edge feature extraction network, which utilizes the rich edge information of herbage to assist the extraction of main feature to improve

**Table 1. Convolutional neural network related.**

| Author | Year | Classification | Dataset | Model | Accuracy |
|---|---|---|---|---|---|
| Shahbaz KhanI [28] | 2021 | 2 | Farm | Developed proposed system | 0.90 |
| Yanlei Xu [29] | 2021 | 12 | PlantSeedlings dataset | Xception [30] | 0.9963 |
| Vo Hoang Trong [31] | 2021 | 12 | PlantSeedlings dataset | MobileNet [32] | 0.9970 |
| Xue Yan [33] | 2020 | 6 | Farm | CNN | 0.945 |
| Kun Hu [34] | 2020 | 8 | DeepWeeds dataset | Graph weeds net | 0.981 |
| Vo Hoang Trong [35] | 2020 | 12 | Cropped plants V2 dataset | CNN | 0.9877 |
| Adnan Farooq [36] | 2019 | 4 | Hyperspectral dataset | CNN | 0.9472 |
| Alex Olsen [37] | 2018 | 8 | DeepWeeds dataset | ResNet50 [38] | 0.957 |
| Mads Dyrmann [39] | 2016 | 22 | Six different datasets | CNN | 0.862 |

the classification accuracy. 3: The pre-trained autoencoder network is employed to assist the main feature extraction network to classify.

## Related work

Traditional machine learning algorithms have been widely used in computer vision [5, 25–27], and achieved good results. We sorted out the current popular machine learning and deep learning methods about forage recognition. Since most of the grass data sets are built for themselves, the experimental results of different algorithms for different data sets can not be compared objectively, the grass recognition technology will be elaborated according to the time line, as shown in Table 1.

### CNN

Vo Hoang Trong et al. [31] proposed a yielding multi-fold training (YMufT) strategy to train a Deep Neural Networks (DNNs) model on an imbalanced dataset. This strategy reduces the bias in training through a min-class-max-bound procedure (MCMB), which divides samples in the training set into multiple folds. The model is consecutively trained on each of these folds. They carried out a series of experiments on several datasets, and the results show that the proposed algorithm has high accuracy. Shahbaz KhanI et al. [28] proposed an optimized semi-supervised learning approach, offering a semi-supervised generative adversarial network for crops and weeds classification at early growth stage. In this method, only a few labeled data can be used to complete the weed classification by using the Generative Adversarial Networks (GAN). The proposed system is evaluated extensively on the Red Green Blue (RGB) images obtained by a quadcopter in two different croplands (pea and strawberry). This method achieves 90% accuracy when 80% of training data is not labeled. It has a remarkable effect for the future development of agricultural intelligence, but the GAN depends on a large number of data sets to generate realistic training data, and for the two classification, 90% accuracy is not high enough.

Yanlei Xu et al. [29] proposed a weed recognition method using depthwise separable CNN based on deep transfer learning. In this method, convolution neural network is used to extract grass features, global pooling is used to replace full connection to extract feature vector, and xgboost is used as classifier instead of full connection layer to complete classification. Although this method achieves high accuracy, xgboost belongs to the older tree algorithm, and the newer machine learning classification algorithms may obtain higher accuracy. Besides, the data set is an unbalanced data set, which affect the generalization ability of the model.

Xue Yan et al. [33] employed traditional machine learning algorithm to extract edge, color and texture for classification, and compared with Deep Convolutional Neural Network (DCNN) algorithm. Results had demonstrated DCNN algorithms obtain the highest accuracy. It can be concluded that the deep learning algorithms have stronger feature extraction ability than traditional algorithms. In addition, the model structure is relatively simple. It can achieve higher accuracy in the recently proposed image classification algorithms.

Kun Hu et al. [34] proposed a novel graph-based deep learning architecture, namely Graph Weeds Net(GWN), which aims to recognize multiple types of weeds from conventional RGB images collected from complex rangelands. The algorithm combines graph convolution neural network, LSTM algorithm and image classification algorithm to complete the classification of eight kinds of weeds [37]. Although the final classification effect is improved, the complexity of the network model is also increased.

Vo Hoang Trong et al. [35] developed a novel classification approach via a voting method by using the late fusion of multimodal DNNs. The score vector used for voting is calculated by Bayesian conditional probability method or determining the priority, so that the score vector obtained from the classification model can get more weight in the final classification. This method can improve the accuracy of the network model, however, the model's complexity will increase exponentially with the increase of the number of model parameters.

Adnan Farooq et al. [36] used patch-based classification approache, CNN and histogram of oriented gradients (HOG) methods are evaluated and compared. It is finally concluded that CNN has better feature extraction ability for images and the accuracy of CNN is more easily influenced by the input image size. Alex Olsen contributes the first large, public, multiclass image dataset of weed species from the Australian rangelands, allowing for the development of robust classification methods to make robotic weed control viable. The DeepWeeds dataset consists of 17,509 labeled images of eight nationally significant weed species native to eight locations across northern Australia. They used the benchmark deep learning algorithm to train and identify the data set, and finally achieved 95.7% accuracy.

Mads Dyrmann et al. [39] presented a method that is capable of recognizing plant species in colour images by using a CNN. By changing light conditions and soil types to increase the complexity of the dataset, 22 kinds of weeds and crops, a total of 10413 images were taken. And 86.2% accuracy is achieved on the network designed by themselves, but the processing time of a single plant is 27 ms, which is not strong for the real-time detection of weeds.

## Traditional machine learning

In Table 2, the related technologies of using machine learning to identify grasses are summarized. Shanwen Zhang [40] proposed a LWMDP based weed recognition method. In this method, Grabcut is firstly used to remove the most background, FMC is utilized to fast segment the weed image, and then LWMDP is employed to reduce the dimension of the segmented weed image, finally SVM is adopted to recognize weed species. This method not only

**Table 2. Traditional machine learning related.**

| Author | Year | Classification | Dataset | Model | Accuracy |
|--------|------|---------------|---------|-------|----------|
| Shanwen Zhang [40] | 2019 | 8 | South China crop field | SVM | 0.9235 |
| Adel Bakhshipour [41] | 2018 | 4 | experimental sugar beet fields | SVM | 0.9967 |
| David Hall [42] | 2015 | 10 | Flavia | Combined system | 0.973 |
| Luís A.M. Pereira [43] | 2012 | 3 | São Paulo State University | MS-SVM-RBF | 0.8715 |
| Asnor Juraiza Ishak [44] | 2009 | 2 | Natural lighting conditions | GFD and GW | 0.94 |

is effective, but also eliminates the interference of background information. However, due to the binarization operation, dimension reduction loses a lot of details of grasses uncontrollably.

Adel Bakhshipou et al. [41] adopted Fourier descriptors and moment invariant features to extract different shape characteristics of weeds. After fusing these several features, they utilized SVM as classifier and obtained the highest accuracy, about 95.00%. This method mainly relies on the shape features of weeds, but ignores the details of weeds, such as texture, color difference and so on. If the useful feature information is ignored, the final classification effect will be reduced.

David Hall et al. [42] used deep convolutional neural network (ConvNet) features and introduced a range of condition variations to explore the robustness of these features, including: translation, scaling, rotation, shading and occlusion. Evaluations on the Flavia dataset demonstrate that in ideal imaging conditions, combining traditional and ConvNet features yields state-of-the-art performance with an average accuracy of 97.3%±0.6%.

Luís A.M. Pereira et al. [43] established the image data set of aquatic weed leaves, applied five kinds of shape analysis and six kinds of supervised machine learning technology for comparative analysis, and finally achieved 96.41% accuracy on the data set by using the Optimumpath Forestsuan algorithm. These two methods have limited effect on the recognition of leaves in the natural environment because they use the leaves collected separately.

Asnor Juraiza Ishak et al. [44] used the combination of Gabor wavelet (GW) and gradient field distribution (GFD) technology to extract a new set of feature vectors for weed classification. The overall classification accuracy is 94% using this technique, while 84% using only the features of GW algorithm. Compared with GW algorithm, GFD algorithm improves the objective accuracy, but it can identify fewer kinds of objects, so it may not be suitable for multiple object data sets.

From the above grass classification dataset and algorithms, it can be seen that for the complex weed dataset, the DCNN algorithm with stronger learning ability is generally used, while for the relatively simple weed data sets, the machine learning algorithms are typically used. Generally, the machine learning algorithm has simple models and is suitable for processing simple low dimensional data. For image data, it is generally necessary to reduce the dimension by binarization, and then use machine learning algorithms for classification. However, the growth environment of weeds is very complex, so using deep convolution neural network to classify them is a more reliable strategy. For DCNNs, the main problem is that with the deepening of the network, the shallow features will disappear inevitably. ResNet [38], a network with residual connections, alleviates this problem to some extent. But inevitably, some features that are helpful to classification will disappear in the process of convolution. To alleviate this problem, we propose an edge autoencoder network. The extracted edge information and overall feature information are enhanced and input to the basic CNN to improve the accuracy of the algorithm.

## Materials and methods

### Overall framework

The overall forage grasses classification method is shown in Fig 1. Firstly, we construct a herbage dataset. Several operations like data enhancement, data equalization and image segmentation are employed to preprocess the collected data. The segmented image is input into the pretrained coding network, backbone feature extraction network and edge extraction network simultaneously. The whole features extracted by autoencoder network and the edge features extracted by edge extraction network are input into the backbone feature extraction network, which are used for classification.

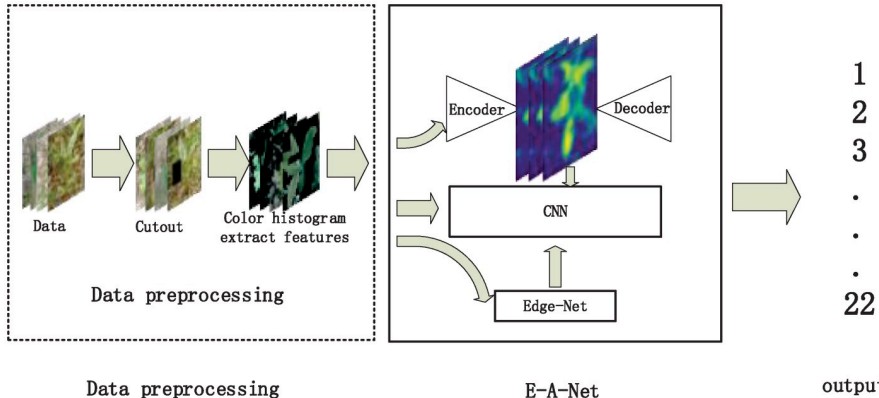

**Fig 1. Forage grasses identification process.**

## Datasets

The forage grasses images used in the experiment were collected from Erdos City, Etuoke Banner, Inner Mongolia Yiwei White Cashmere Goat Co, Ltd. and No.1 Branch factory. The longitude and latitude of data acquisition are $39°10'25.8420''$N and $107°54'35.8380''$E. The forage grasses data were collected in two times and photographed on June 17, 2020 and October 7, 2020. Due to the shortage of forage grasses collected for the first time, the forage grasses dataset was supplemented and enriched for the second time. The photos were taken by mobile phone. The photo pixel was $4000 \times 3000$, the photo format was RGB, and the shooting position was 10cm-40cm away from the forage grasses. A total of 3889 photos were collected. Their types were as follows Fig 2:

## Data preprocessing

The resolution of the original images is $4000 \times 3000$. Since high resolution will lead to a slow training process and consume a lot of GPU resources, the dataset is resized to $224 \times 224$. In the process of photographing and collecting the images, the sample data has a long tail phenomenon due to the scarcity of some forage grasses species. In order to alleviate the impact of this phenomenon on the training process, we carry out some operations to balance data. The distribution of forage grasses species is statistically visualized in the blue histogram in Fig 3 below: it can be seen that the number of forage grasses species is seriously uneven [8]. In this paper, the cutout data enhancement method is adopted [7]. It can not only enhance the dataset, achieve the purpose of balancing the dataset, but also improve the generalization ability of the network model [8]. The number of each species after data enhancement is shown in the yellow histogram as Fig 3.

After the model is trained, the class activation thermogram [45] is visualized. It is found that because forage grasses usually have different backgrounds, CNN can identify forage grasses species rather than forage grasses information from the image background for some kinds of forage grasses, which decreases the generalization ability of the network model. In order to solve this problem, image segmentation algorithm based on the threshold value is adopted. Since illumination has a strong impact on the image recognition in outdoor conditions and the HSV color space is robust to the change of illumination, it is more suitable to describe the color image information under the change of illumination. Therefore, firstly, the RGB image is converted to HSV color space, and the grass and background image is separated

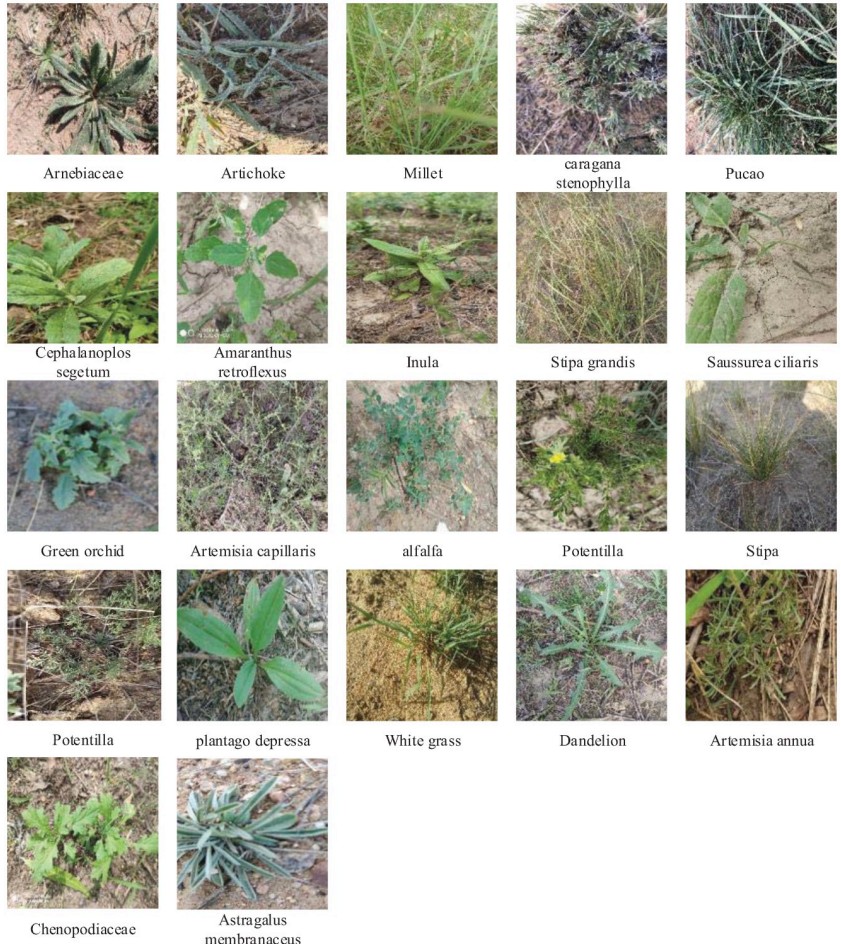

**Fig 2. Photos of herbage species.**

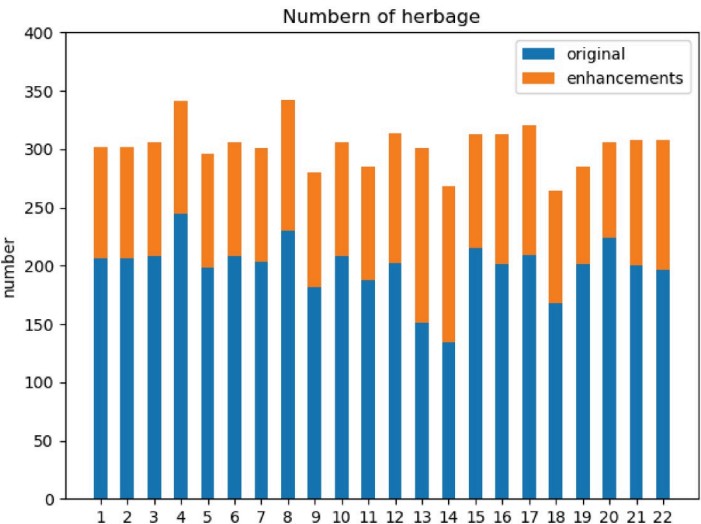

**Fig 3. Quantity before and after data balance for each type of forage grasses.**

by segmentation method based on a threshold value [6]. Afterwards, the processed herbage dataset is randomly divided into 60% training set, 20% validation set and 20% test, respectively.

## Method

A novel edge autoencoder network is proposed in this study. More concretely, the algorithm is divided into three parts: the first part is the backbone feature extraction network, the second part is the edge extraction network based on a Sobel operator, and the third part is the pre-trained autoencoder network. The edge extraction network based on a Sobel operator is mainly used to extract the edge texture information of forage grasses [12]. The residual connection between the edge information and the characteristic information of forage grasses extracted from the main network plays a role in preventing the edge information of forage grasses from disappearing in the deep network. The pre-trained autoencoder network is used to fix the weight of the autoencoder network during the overall network training [20–22] and obtain the overall characteristics of forage grasses through the coding part. This operation guarantees the network pays attention to the overall characteristics of forage grasses, which forms a hard attention mechanism to improve the network's ability of memorizing the overall characteristics as well as the classification accuracy. The overall structure of E-A-Net is shown in Fig 4.

**The network part of edge extraction.** Edges are the basic features of an image, and they enrich characteristic information. They often exist between target and background, which is not only important for image segmentation [46] but also useful for image classification [12, 14] and target detection [47]. Based on this, the edge extraction network containing Sobel

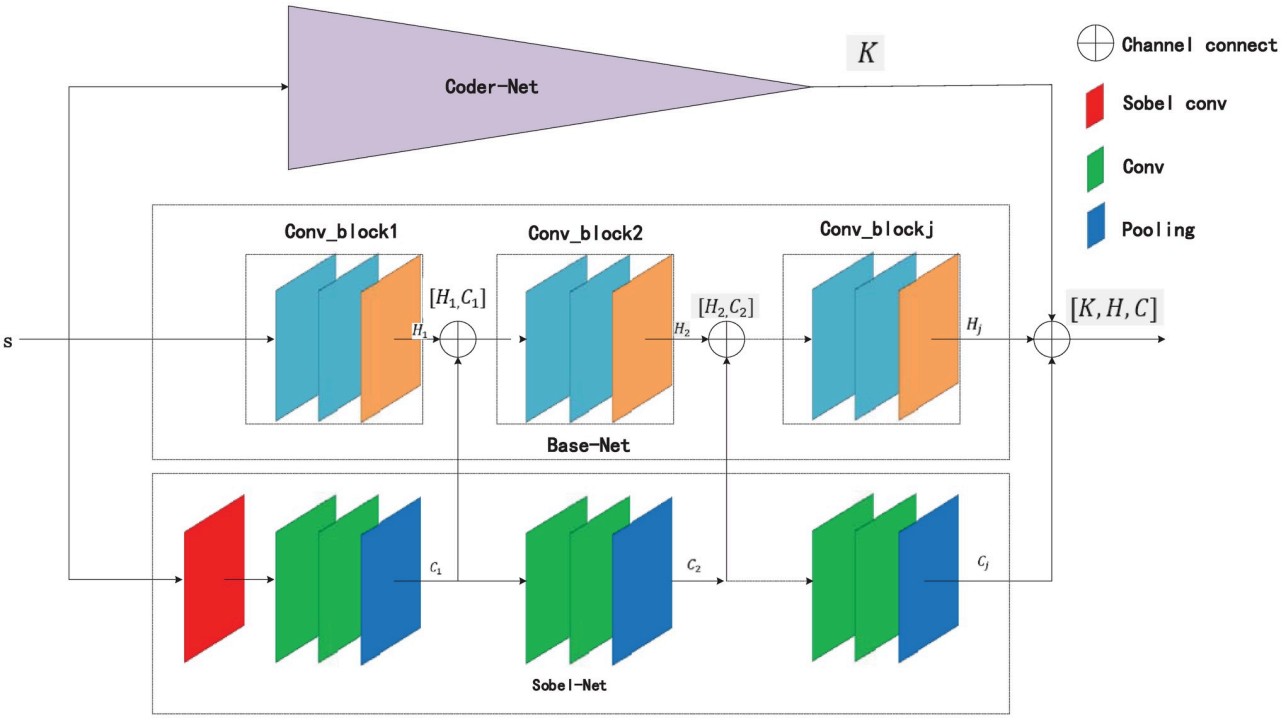

**Fig 4. E-A-Net network structure.**

operator is added to the backbone feature extraction network to focus on extracting edge information.

In this paper, four Sobel operator modules with 3 rows and 3 columns are used to detect the size of the pixel values of the feature map in 45˚, 90˚, 180˚, 315˚ edge direction respectively. The convolutional results of the Sobel operator and the feature map can be calculated as:

$$f_z = s \otimes q \tag{1}$$

The pixel value of the feature map $f_z$ obtained by convolution of four Sobel operators in different directions is used to obtain the feature map. $s$ is the original image, and $q$ is the convolution kernel of Sobel operator. $z$ represents four convolution kernels in different directions. After five maxpooling operations, the size of the feature map is sampled from $224 \times 224$ to $7 \times 7$. The feature map and the feature map with the same size as the backbone network are spliced in the channel direction, and the extracted multi-scale edge feature information is fused into the backbone feature extraction network [24, 47].

**The network part of the autoencoder network.** Unlike the previous Squeeze-and-Excitation Networks [15] and Convolutional Block Attention Module (CBAM) [16], autoencoder network acquires attention mechanisms in a way that is not limited to inter-channel or inter-pixel attention, but uses the overall characteristic of an object as an attention mechanism. Based on this, the embedded layer of the autoencoder network is fused into the backbone CNN to provide a focus on the characteristics of the object itself. The detailed structure of the autoencoder network is show in Fig 5.

The autoencoder network selected in this paper uses the modified structure of U-Net [23]. In the coding part, the network adopts a convolutional layer with a convolutional kernel size of $4 \times 4$, a step size of 2 and a Relu activation function instead of the original convolutional pooling layer reference. In the decoding part, the coded vector is deconvoluted with a convolutional kernel size of $4 \times 4$ and a step size of 2 to reconstruct the image features, and then these features are fused with a $1 \times 1$ convolution [48]. The final output of the network is regarded as the decoded image.

Suppose that the dimension of input layer and output layer of the autoencoder network is $n$, the dimension of hidden layer is $m$, and the number of samples is $N$, then the sample set is:

$$S = \{x_i\}_{i=1}^N \tag{2}$$

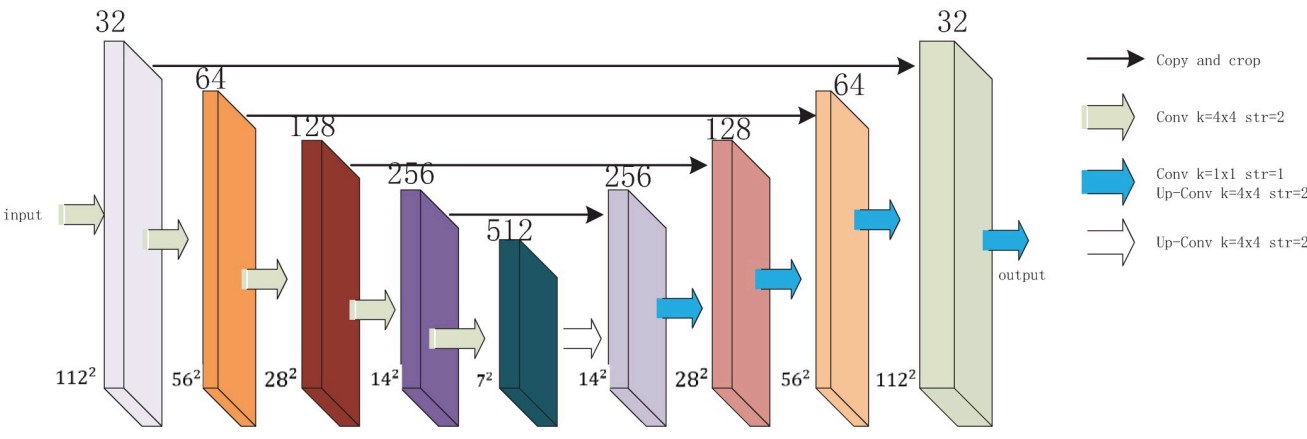

**Fig 5. Detailed structure of autoencoder network.**

The autoencoder is a network that reconstructs the input signal $x$ with convolution operation, encodes it into a high-dimensional vector, decodes and restores the original input signal $x$ according to the high-dimensional vector. The encoder network and decoder network in the above figure are both composed of the convolutional layers, which transform the input signal into an abstract high-dimensional vector $y$. The decoder network decodes the input signal into $k$, which is similar to the original signal according to the abstracted $f$. Suppose that the encoding network function is $f$, the decoding network function is $g$, the weight matrix for linear transformation is $w$, and the offset is $b$. Then the output of the encoder is:

$$k = f(x) = \sum_{i=1}^{n} f(w_i x_i + b) \tag{3}$$

The linear element Relu is chosen as the activation function of the encoder. Because it has the characteristic of sparse activation, and the derivative of input part greater than zero is constant, Relu will not cause gradient dispersion.

The decoder is the reverse operation of the encoder, and the function of the encoder is to reconstruct the intermediate vector. The function expression of the decoder:

$$y = g(k) = \sum_{i=1}^{N} g(\tilde{w} f(x_i) + b) \tag{4}$$

The input data $x$ is predicted to obtain the output data $y$. If the output data $y$ is basically similar to the input data $x$, then autoencoder will retain the important information of most signals. Therefore, the autoencoder network is considered well trained. In general, the mean square error function is used to measure the difference between the input data $x$ and the output data $y$:

$$L(x, y) = MSELoss(x, y) = \frac{1}{n} \sum_{i=1}^{n} (x_i - y_i)^2 \tag{5}$$

When the training sample is $S$, the overall loss function of the autoencoder network is as follows:

$$J_{autoencoder}(\theta) = \sum_{x \in S} L(x, g(k)) \tag{6}$$

Finally, the loss of the CNN is iteratively calculated by gradient descent algorithm, so that the parameters $x$ and $b$ of the autoencoder network can be continuously updated through back-propagation, which is completed by the training of the autoencoder network.

Because the original image can be reconstructed by the network according to the abstracted features, it indicates that the encoded $y$ has all the important information of the original signal $x$. Therefore, it is effective to use this information as the classification basis of CNN. The network designed in this paper is to splice the feature map of CNN before the full connection layer and the middle feature layer of the autoencoder network to utilize the effective feature information.

**The whole algorithm part.**   The image $x$ with the size of (224, 224, 3) is input into the trained autoencoder network, edge extraction network and backbone CNN in turn. Let the first $J$ functions of the edge detection network be $M_1, M_2, \ldots \ldots M_J$. When the image $s$ passes through the edge extraction network, the feature information is extracted respectively $C_1, C_2, \ldots \ldots C_J$:

$$C_1 = \sum_{i=1}^{n} M_1(w_{1i} x_i + b) \tag{7}$$

$$C_2 = \sum_{i=1}^{n} M_2(w_{2i} C_1 + b) \tag{8}$$

$$\ldots\ldots\ldots$$

$$C_J = \sum_{i=1}^{n} M_n(w_{Ji} C_{n-1} + b) \tag{9}$$

The first $j$ functions of the edge detection network are represented as $G_1, G_2, \ldots\ldots G_j$. These extracted features are spliced with other extracted features on the channel, and the latter features come from the first $j$ convolutions of the backbone CNN: $H_1, H_2, \ldots\ldots H_j$, the channel splicing operation is represented by [].

$$H_1 = \sum_{i=1}^{n} G_1(w_{1i} x_i + b) \tag{10}$$

$$H_2 = \sum_{i=1}^{n} G_2(w_{2i}[H_1, C_1] + b) \tag{11}$$

$$\ldots\ldots\ldots$$

$$H_{j-1} = \sum_{i=1}^{n} G_{j-1}(w_{(j-1)i}[H_{j-2}, C_{j-2}] + b) \tag{12}$$

$$H_j = \sum_{i=1}^{n} G_j(w_{ji}[H_{j-1}, C_{j-1}] + b) \tag{13}$$

The outputs of three networks, $K, C_j, H_j$ are used to splice the feature map in channel dimension to obtain feature extracted from multiple networks. Obtained feature is input into $1 \times 1$ convolution for information fusion between channels to obtain the final network output.

## Results and discussion

### Experimental equipment

The software platform used in this experiment is anaconda3, with Tensorflow and keras frameworks. The processor is Intel Core i5 9300CPU @2.4GHz. RAM size is 16, operating system is 64-bit, graphics card model is NVIDIA GeForce GTX1660Ti, and video memory size is 6G.

### Data processing

GAN [49] is tried to enhance the dataset. Due to the small amount of data, the resulting photos are very similar and low quality. The cutout is used as an alternative to enhance the dataset. The acquired image after data enhancement is shown in Fig 6. Some characteristics of the forage grasses are obscured, which is helpful for network to learn more about the characteristics forage grasses, not just one part. So cutout [7] ultimately not only balances dataset but also improves the generalization ability of the network [10].

### Pre-training of autoencoder network

The size of the image used in the autoencoder network pre-training process is $224 \times 224$. The normalization operation is performed to prevent the gradient explosion [38] in the training process, which makes the model difficult to fit. A total of 500 epochs of autoencoder network pre-training are performed. The settings used are as follows: the image batch size is 32, the network optimizer is Adam [50], the learning rate is 0.0001, and the tensorboard is used to visualize the weight changes during the network training process.

It can be seen from Fig 7 that the loss value of autoencoder network in training set and validation set decreased sharply in the first 50 rounds of training, and decreased slowly after 100 rounds. The loss value of the final validation set reached 0.003721473. After 500 epochs of

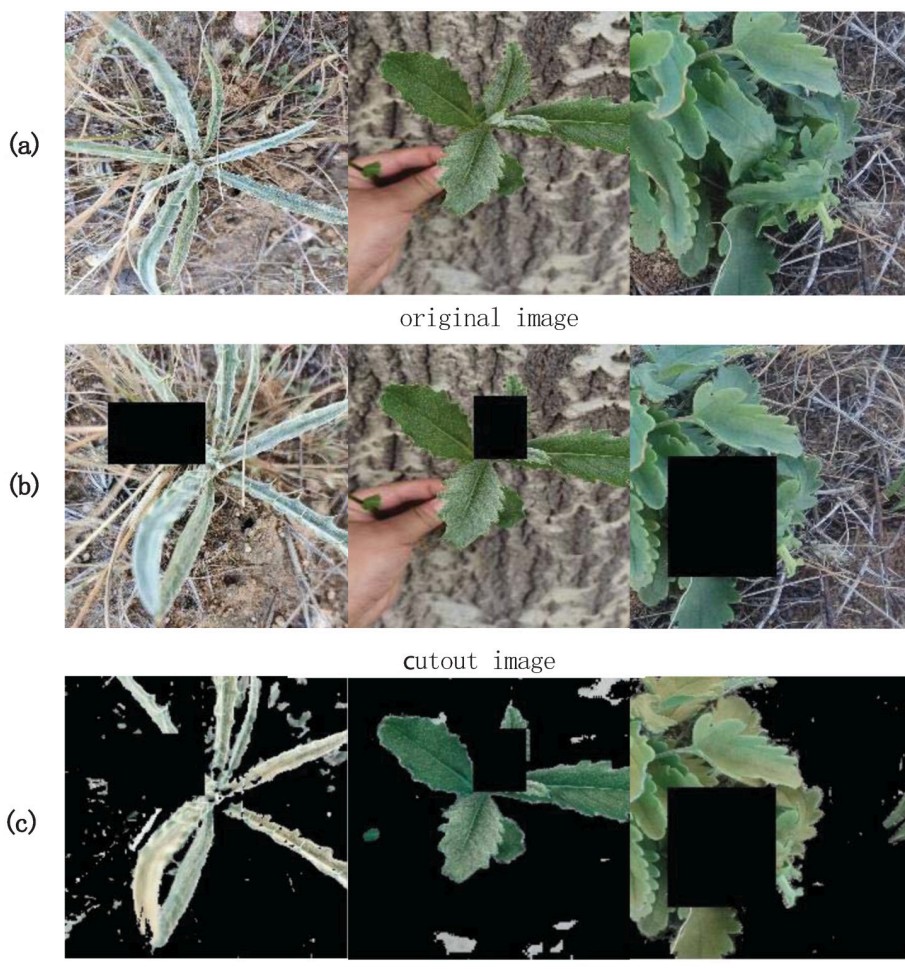

**Fig 6. Data processing results: The first row is the original data, the second is the cutout enhanced data, and the third is the background segmentation data.**

training, the autoencoder network model can restore the original image well. The weight file corresponding to the minimum loss value is selected to reconstruct the autoencoder network.

After feeding the processed dataset into the network model, the class thermogram is activated [45]. The thermogram is shown in Fig 8. Because different forage grasses may exist in different growing environments and their backgrounds are not well used, the CNN learns more about the background information of this part of the forage grasses pictures in the classification process, thus the true information of the forage grasses is ignored, and the generalization ability of the trained model is poor. Image segmentation algorithm based on a threshold value is used to extract herbage features, remove background information and make the network model more focused on learning herbage characteristics. The image of the dataset after cutout and background segmentation is shown in Fig 8 below:

## Training strategy of E-A-Net

Firstly, the size of the data is adjusted to 224 × 224 for E-A-Net. Then, the resized data is augmented and normalized. After that, the input data is fed to the autoencoder network, backbone

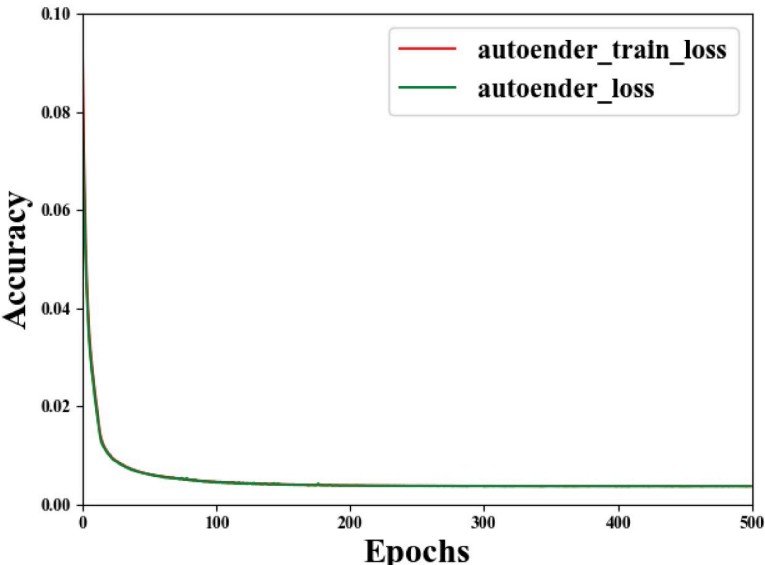

**Fig 7. Loss curve of autoencoder network.**

feature extraction network and edge extraction network at the same time. In the training process of E-A-Net, extra special operations are needed by the autoencoder network, to be more specific, when entering into the autoencoder network, the network needs to read the autoencoder weights that have been trained separately, the autoencoder network is freezing, and it only needs to obtain the embedded layer feature map that is part of the output of the encoding and inputs it into the main feature extraction network. Finally, the network is trained for 200

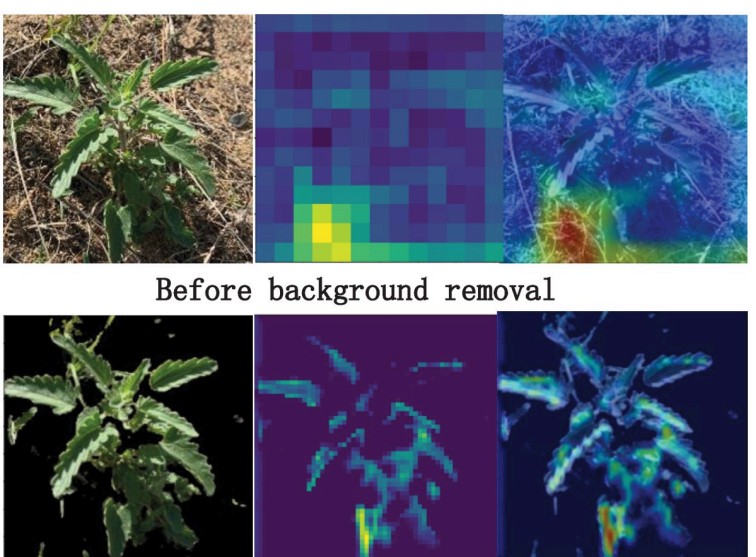

**Fig 8. Comparison of active thermograms before and after background removal.**

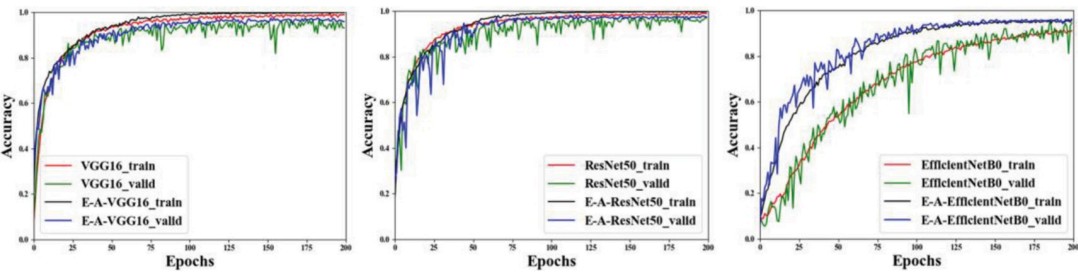

**Fig 9. Training validation curve.**

epochs with a batch size of 4. The used optimizer is Adam [50] and the learning rate is 0.0001. The tenorborad is also used to monitor the weight change and the characteristic map transformation of the convolutional layer in the network training process in real-time.

## Training result of E-A-Net

The experiment results are compared with three baseline networks, VGG16, ResNet50 and EfficientNetB0 and improved E-A-VGG, E-A-ResNet50 and E-A-EfficientNetB0.

As can be seen from Fig 9, E-A-VGG16 and E-A-ResNet50 are faster than VGG16 [51] and ResNet50 [38] in learning speed. The process of E-A-EfficientNetB0 [52] training network on a forage grasses dataset is more accurate both training set and the validation set than in the original classification network, and the result curve is more stable in the validation set.

The above 12 curves can more accurately show the accuracy changes of different models during the training process. We can see that after 100 rounds the accuracy waveform of the network model tends to be smooth and rises slowly. Therefore, we select the last 100 accuracy values and 7 indicators to study the performance of the algorithm. The train and validation results analysis are as Table 3:

In training set, E-A-VGG16 has the highest accuracy compared with other algorithms, 0.998856. EfficientNetB0 has the worst performance, 0.914344. Compared with other algorithms, E-A-ResNet5 has the minimum standard deviation at 0.001631, while EfficientNetB0 has the maximum standard deviation at 0.038122. Therefore, it can be concluded that E-A-ResNet50 has a gentle learning and small fluctuation in the training set. This conclusion can

**Table 3. Train and validation results analysis.**

|  | Count | Mean | std | Median | Min | Max | Difference |
|---|---|---|---|---|---|---|---|
| VGG16 [51] train | 100 | 0.981587 | 0.005450 | 0.982298 | 0.965738 | 0.992691 | 0.011104 |
| VGG16 [51] valid | 100 | 0.938200 | 0.021919 | 0.941204 | 0.817593 | 0.992691 | 0.032170 |
| E-A-VGG16 train | 100 | 0.995430 | 0.002451 | 0.996110 | 0.984668 | 0.992691 | 0.003426 |
| E-A-VGG16 valid | 100 | 0.962222 | 0.006415 | 0.962963 | 0.943519 | 0.992691 | 0.012778 |
| ResNet50 [38] train | 100 | 0.982360 | 0.005039 | 0.983212 | 0.969849 | 0.992691 | 0.011244 |
| ResNet50 [38] valid | 100 | 0.951850 | 0.018886 | 0.956481 | 0.875000 | 0.992691 | 0.031483 |
| E-A-ResNet50 train | 100 | 0.995386 | 0.001631 | 0.996574 | 0.991092 | 0.992691 | 0.003015 |
| E-A-ResNet50 valid | 100 | 0.971361 | 0.003905 | 0.971296 | 0.961111 | 0.992691 | 0.010119 |
| EfficientNetB0 [52] train | 100 | 0.860690 | 0.038122 | 0.870375 | 0.775468 | 0.992691 | 0.053654 |
| EfficientNetB0 [52] valid | 100 | 0.866204 | 0.045947 | 0.870833 | 0.719444 | 0.992691 | 0.089352 |
| E-A-EfficientNetB0 train | 100 | 0.940998 | 0.013105 | 0.945523 | 0.897899 | 0.992691 | 0.017431 |
| E-A-EfficientNetB0 valid | 100 | 0.947398 | 0.010777 | 0.950463 | 0.911111 | 0.992691 | 0.014639 |

also be verified by the waveform fluctuation of E-A- ResNet50 compared with other algorithms. Meanwhile, according to the maximum and median, we can estimate the accuracy improvement of the network model after 150 rounds. We can draw the conclusion that the accuracy improvement of EfficientNetB0 is up to 0.053654, and the improvement of E-A-ResNet50 is at least 0.003015. This shows E-A-ResNet50 has strong learning ability.

In validation set, E-A-ResNet50 has the highest accuracy compared with other algorithms, 0.981481. EfficientNetB0 has the worst performance, 0.955556. Compared with other algorithms, E-A-ResNet50 has the minimum standard deviation with 0.003905 while EfficientNetB0 has the maximum standard deviation with 0.045947. It can be seen that E-A-ResNet50 has stable effect and high accuracy in the validation set. According to median and maximum, the accuracy improvement of network model after 150 rounds can be estimated. The accuracy of E-A-ResNet50 is 0.010119, while that of EfficientNetB0 is the highest, 0.089352. In short, we can get a consistent conclusion: E-A-ResNet50 has a strong learning ability, and in the first 150 rounds of training, it has basically completed the parameter learning.

Meanwhile, we also find a phenomenon which is inconsistent with the other two models. For EfficientNetB0 and E-A-EfficientNetB0, their accuracy in the validation set is generally higher than that in the training set, which proves that the algorithm has good generalization ability. Compared with the other four algorithms, these four algorithms' accuracy are generally higher in the training set than that in the validation set, which indicates that these four algorithms have slight overfitting. Therefore, we choose the most accurate weight in the validation set to make the final evaluation of the above six models on the test set.

From the visual feature map of the edge extraction network in Fig 10, it is observed that the edge extraction network extracts and retains the grass edge features very well, and the pixels of the edge part have higher values. The class activation feature map [45] shows that the edge information extracted by the edge extraction network plays an important role in the final E-A-Net classification process.

From Fig 11(b), it can be seen that most of the main characteristics of forage grasses are retained by the autoencoding network. The stitching of the channels with the backbone feature extraction network can make up for the deficiency that most of the neural networks have blank feature maps on the feature channels of the deep convolutional layer due to the deepening of the network level. From Fig 11(c) and 11(d), it is concluded that the main network model with an edge extraction network is more widely involved in the final classification decision.

## Test result of E-A-Net

The test result are shown in Fig 12. Restults show that E-A-ResNet50 improves the classification accuracy of forage grasses 15 from 25% to 95%, E-A-EfficientNetB0 improves the classification accuracy of forage grasses 17 from 39% to 100%. It can be seen that the original E-A-Net has better learning ability than the original convolutional network and can effectively avoid the problem of poor recognition accuracy of a certain forage grasses.

To measure a model performance in a comprehensive way, the five indicators of the test set are used to valuate model performance: *precision*, *recall*, *f1 − score*, *params* and *cost*. The test results are as Table 4:

*Precision*: refer to the proportion of positive samples in positive examples determined by the classifier.

$$\mathrm{pr}ecision = \frac{TP}{TP + FP} \tag{14}$$

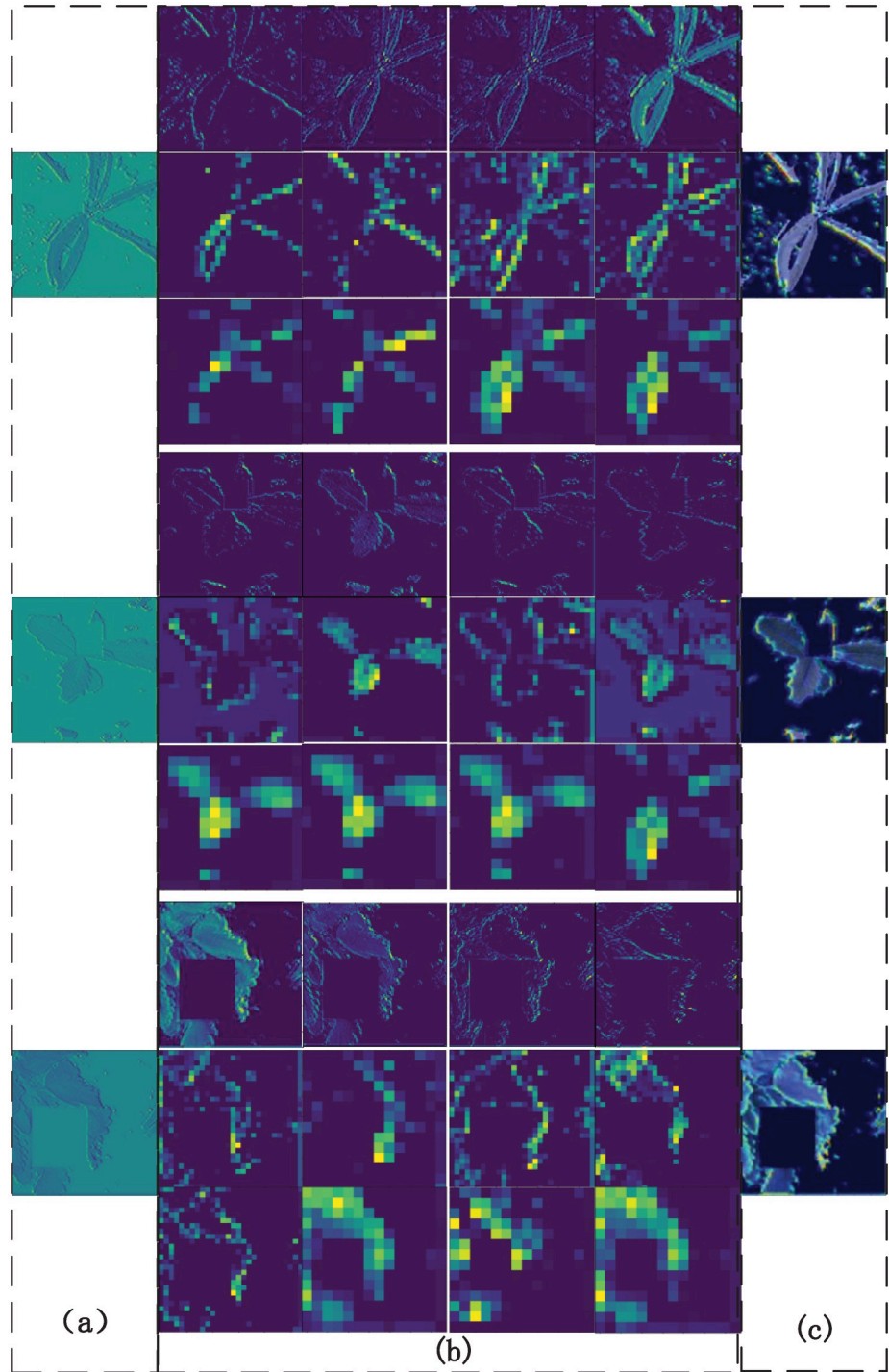

**Fig 10. The influence of edge information on model recognition.** (a) Feature maps output by a Sobel operator, (b) Feature maps of the first three pooling layers of the network extracted for edges, (c) Extracting network feature map for class-activated edges.

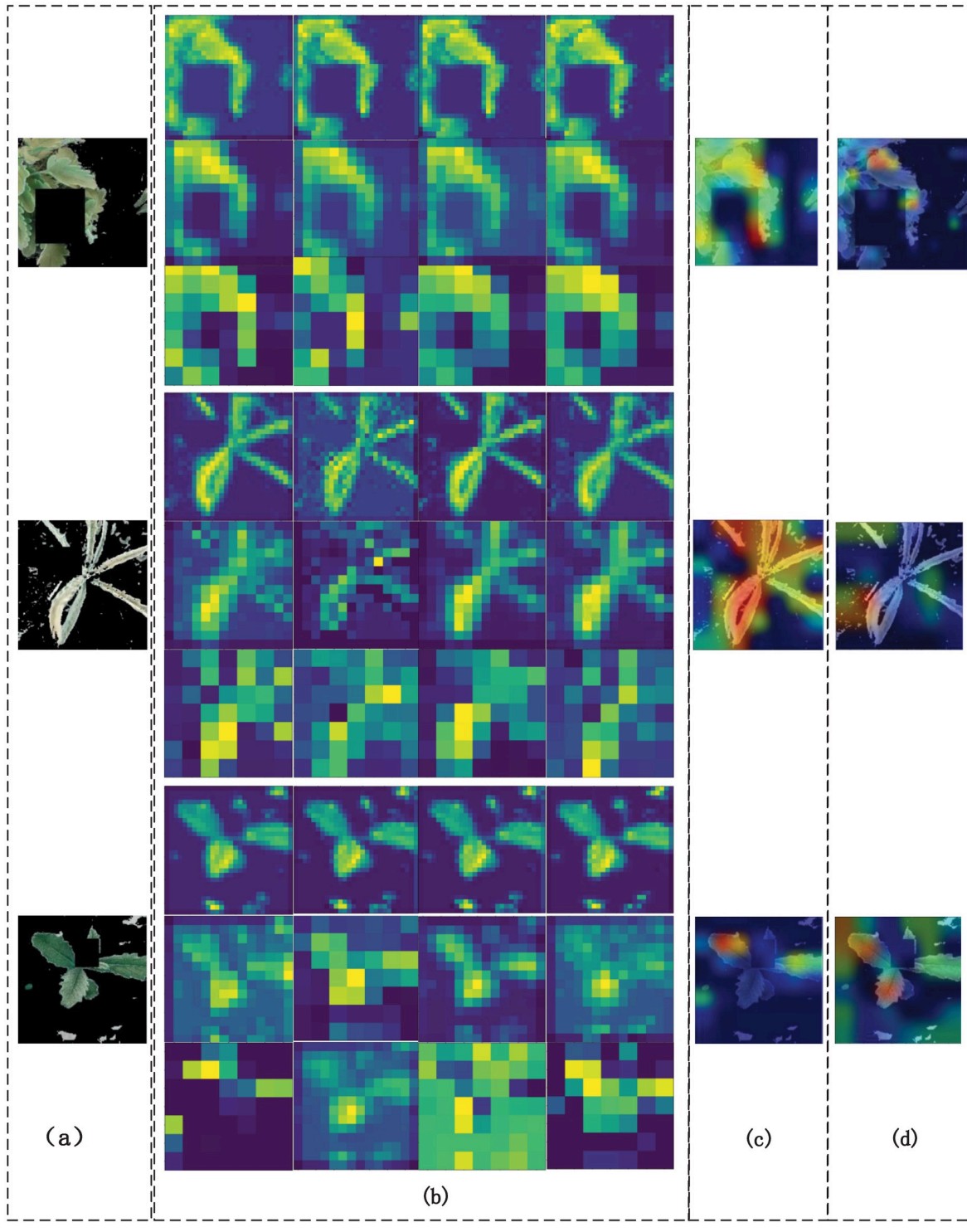

**Fig 11. The influence of autoencoding network on the overall model.** (a) Image input to autoencoder network, (b) Result of last two convolutional layers and last pooling layer of autoencoder network, (c) Class activation E-A-Net autoencoder network partial characteristic diagram, (d) Class activation basic neural network thermogram.

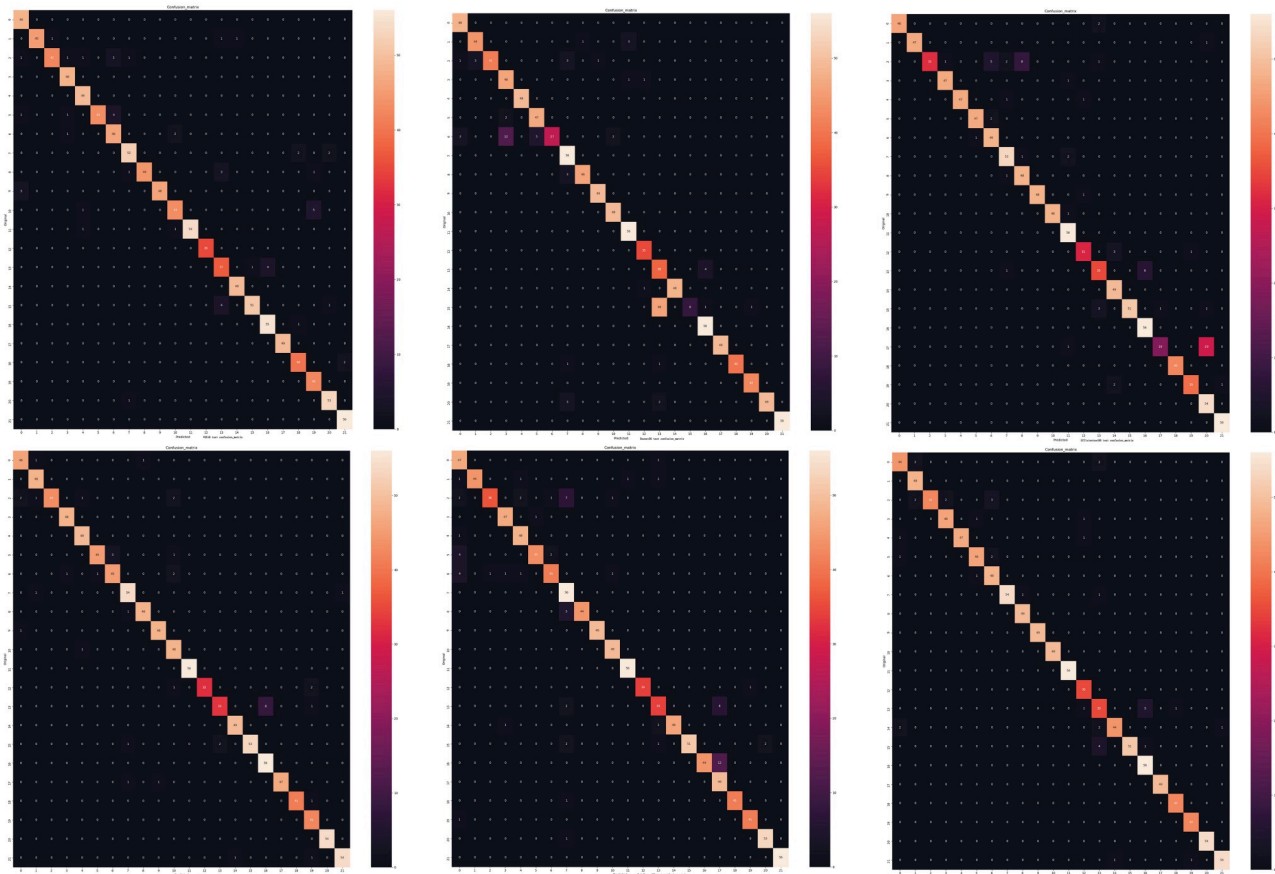

**Fig 12. Confusion matrix diagram: Where the horizontal axis of the confusion matrix is the real type and the vertical axis is the prediction type.**
On the first line is the confusion matrix for VGG16 [51], ResNet50 [38], and EfficientNetB0 [52] networks, and on the second line is the confusion matrix for E-A-VGG16 and E-A-ResNet50, E-A-EfficientNetB0 networks.

*Recall*: refer to the ratio of predicted positive cases to the total positive cases.

$$recall = \frac{TP}{TP + FN} \tag{15}$$

**Table 4. Test result.**

| | precision | recall | f1-score | params | cost |
|---|---|---|---|---|---|
| VGG16-Backbone | | | | | |
| VGG16 [51] | 0.95 | 0.95 | 0.95 | $14.7 \times 10^6$ | 0.0457s |
| E-A-VGG16 | 0.96 | 0.96 | 0.96 | $18.1 \times 10^6$ | 0.0502S |
| ResNet50-Backbone | | | | | |
| ResNet50 [38] | 0.93 | 0.91 | 0.91 | $23.6 \times 10^6$ | 0.0521s |
| E-A-ResNet50 | 0.94 | 0.93 | 0.94 | $39.7 \times 10^6$ | 0.0549s |
| EfficientNetB0-Backbone | | | | | |
| EfficientNetB0 [52] | 0.94 | 0.92 | 0.93 | $73.2 \times 10^5$ | 0.0453s |
| E-A-EfficientNetB0 | 0.96 | 0.96 | 0.96 | $85.3 \times 10^5$ | 0.0481s |

*F*1 − *score*: it is the harmonic average of precision rate and recall rate, with the maximum of 1 and the minimum of 0.

$$f1 - score = \frac{1}{n} \sum_{i=1}^{n} \frac{P_i \cdot R_i}{P_i + R_i} \tag{16}$$

*Params* is a standard to measure the amount of network model parameters.

*Cost* is the time consumed to predict a picture:

$$\cos t = \frac{T_n}{n} \tag{17}$$

*F*1 − *score* is used as the final standard to evaluate the model performance. E-A-VGG16, E-A-ResNet50 and E-A-EfficientNetB0 are increased by 1.6%, 2.8% and 3.7% respectively than VGG16 [51], ReNet-50 [38] and EfficientNetB0 [52]. E-A-VGG16, E-A-Resnet50 and E-A-EfficientnetB0 have more $3.4 \times 10^6$, $16.1 \times 10^6$, $12.1 \times 10^5$ parameters than VGG16, ReNet-50 and EfficientNetB0 respectively. In terms of the processing speed for a single image, E-A-VGG16, E-A-ResNet50 and E-A-EfficientNetB0 than VGG16, ReNet-50 and EfficientNetB0 increased by 0.0045*s*, 0.00528*s*, 0.0028*s* respectively. Results show that although there are more parameters in these three improved networks, the accuracy is greatly improved, especially in the EfficientNetB0. Only $1.21 \times 10^6$ parameters and 0.0028*s* of single image operation time are added, but the test accuracy is improved by 3.7%. Therefore, the fusion of the features extracted from the edge extraction network and the features extracted from the autoencoder network into the main features extraction network can effectively improve the accuracy of the network model and the generalization ability.

From the above experimental results, it can be concluded that although E-A-EfficientNetB0 has lower accuracy in the training set and verification set than that of E-A-VGG16, E-A-ResNet50, VGG16 and ResNet50, in test set it has the same effect as E-A-VGG16, and is higher than the other four algorithms in each index. Compared with E-A-VGG16, E-A-EfficientNetB0 has fewer parameters and shorter processing time for a single image. Therefore, E-A-EfficientNetB0 has better performance on forage data sets, which proves the effectiveness of our proposed edge autoencoder network.

## Conclusion and future work

### Conclusion

To sum up, the following conclusions are drawn from this study:

1. An edge autoencoder classification network is proposed. It combines the rich and significant edge information of herbage with the herbage features extracted by the autoencoder network. The proposed method is tested against the herbage dataset of forage grasses in Inner Mongolia and Etuoke Banner. Restults show that the proposed method outperforms the existing benchmarking models. This network can prevent edge features and global features from disappearing with the deepening of network depth.

2. Segmentation of herbage image with complex background can effectively make the network pay more attention to the characteristics of herbage.

3. The number of parameters are increased in the autoencoder network and edge extraction network, which decreases the processing speed of the network model. In the future, the network will be integrated into wearable devices for sheep. Knowledge distillation, pruning

and quantitative algorithms will be explored to further optimize the model, so as to improve its speed.

## Future work

Forage recognition technology can be used in intelligent animal husbandry projects in the future. Therefore, there are still many aspects to be improved in this experiment, and there is still much work to be done in the future. I will list some areas that can be improved in the future.

1. Expansion of dataset: the leaf texture, shape and color of herbage are not exactly the same even for the same kind of herbage. Besides, the color of herbage in May is light green, that in August is dark green, and that in October is a little yellow. Therefore, in order to obtain a more robust herbage recognition model, it is an important, lasting and time-consuming work to expand the forage dataset.

2. Model simplification: our algorithm improves the accuracy of the model to a certain extent, but also increases model's parameters. Knowledge distillation can transfer knowledge from one network to another. These two networks can be isomorphic or heterogeneous. It can be used to transform a network from a large network to a small network, and keep the performance close to that of a large network. In the future, we will design a simplified network model so as to increase the reasoning speed of the network model on the premise of ensuring the accuracy.

3. Production of wearable devices: for the purpose of accurately counting the amount and species of forage, we will make a wearable forage recognition device in the future. According to the sound sensor, we can use the chewing signal of cattle and sheep to control the camera. When the chewing signal is found, we can turn on the camera to obtain the species of forage. The chewing signal is used to count the number and species of forage.

## Author Contributions

**Data curation:** Shilong Zhang, Yushuang Ji.

**Validation:** Xinyu Du.

**Writing – original draft:** Minghua Tian.

**Writing – review & editing:** Ding Han, Caili Gong, Yongfeng Wei, Liang Chen.

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
