## [Decision Letter · Decision Letter 0]

6 May 2021

PONE-D-21-09952

Image classification of forage grasses on Etuoke Banner using edge autoencoder network

PLOS ONE

Dear Dr. Wei,

Thank you for submitting your manuscript to PLOS ONE. After careful consideration, we feel that it has merit but does not fully meet PLOS ONE’s publication criteria as it currently stands. Therefore, we invite you to submit a revised version of the manuscript that addresses the points raised during the review process.

Based on the comments received from the reviewers and my own observation, I suggest major revision for this article.

We look forward to receiving your revised manuscript.

Kind regards,

Thippa Reddy Gadekallu

Academic Editor

PLOS ONE

Journal Requirements:

In your Methods section, please provide additional information regarding the permits you obtained for the work. Please ensure you have included the full name of the authority that approved the field site access and, if no permits were required, a brief statement explaining why.""

In your Methods section, please provide additional location information, including geographic coordinates for the data set if available.

We note that Figure 2 in your submission contain map images which may be copyrighted. All PLOS content is published under the Creative Commons Attribution License (CC BY 4.0), which means that the manuscript, images, and Supporting Information files will be freely available online, and any third party is permitted to access, download, copy, distribute, and use these materials in any way, even commercially, with proper attribution. For these reasons, we cannot publish previously copyrighted maps or satellite images created using proprietary data, such as Google software (Google Maps, Street View, and Earth). For more information, see our copyright guidelines: http://journals.plos.org/plosone/s/licenses-and-copyright.

4a, You may seek permission from the original copyright holder of Figure 2 to publish the content specifically under the CC BY 4.0 license. 

4b, If you are unable to obtain permission from the original copyright holder to publish these figures under the CC BY 4.0 license or if the copyright holder’s requirements are incompatible with the CC BY 4.0 license, please either i) remove the figure or ii) supply a replacement figure that complies with the CC BY 4.0 license. Please check copyright information on all replacement figures and update the figure caption with source information. If applicable, please specify in the figure caption text when a figure is similar but not identical to the original image and is therefore for illustrative purposes only.

Reviewers' comments:

Reviewer's Responses to Questions

**Comments to the Author**

1. Is the manuscript technically sound, and do the data support the conclusions?

Reviewer #1: Yes

Reviewer #2: Yes

2. Has the statistical analysis been performed appropriately and rigorously? 

Reviewer #1: No

Reviewer #2: Yes

3. Have the authors made all data underlying the findings in their manuscript fully available?

Reviewer #1: Yes

Reviewer #2: Yes

4. Is the manuscript presented in an intelligible fashion and written in standard English?

Reviewer #1: Yes

Reviewer #2: Yes

5. Review Comments to the Author

Reviewer #1: 1. In the abstract, the background knowledge on the problem addressed need to be added.

2. In the abstract, the wide range of applications and its possible solutions need to be added.

3. In the abstract, the problem addressed need to be justified with more details.

4. In the Introduction section, the drawbacks of each conventional technique should be described clearly.

5. Introduction section can be extended to add the issues in the context of the existing work

6. Literature review techniques have to be strengthened by including the issues in the current system and how the author proposes to overcome the same.

7. What is the motivation of the proposed work?

8. Research gaps, objectives of the proposed work should be clearly justified.

9. The authors should consider more recent research done in the field of their study (especially in the years 2018 and 2020 onwards).

10. An error and statistical analysis of data should be performed.

11. The conclusion should state scope for future work.

12. Discuss the future plans with respect to the research state of progress and its limitations.

Reviewer #2: 1. What are the limitations of the existing works?

2. List out the main contributions of the current work.

3. Some of the recent works on ML/AI such as the following can be discussed in the paper: "Image-Based malware classification using ensemble of CNN architectures (IMCEC), A Novel PCA-Whale Optimization based Deep Neural Network model for Classification of Tomato Plant Diseases using GPU, Deep learning and medical image processing for coronavirus (COVID-19) pandemic: A survey, Hand gesture classification using a novel CNN-crow search algorithm".

4. Summarize the related works section in the form of a table.

5. Compare the current work with recent state-of-the-art.

6. Present a detailed analysis on the results obtained.

7. Discuss about the limitations of the current work in conclusion.

6. PLOS authors have the option to publish the peer review history of their article (what does this mean?). If published, this will include your full peer review and any attached files.

Reviewer #1: No

Reviewer #2: No

---

## [Author Response · Author response to Decision Letter 0]

29 Jun 2021

Editor : I have incorporated all of your suggestions into my revision. They were very helpful. Thank you.

Reviewer 1: I have incorporated all of your suggestions into my revision. They were very helpful. Thank you.

Reviewer 2: I have incorporated all of your suggestions into my revision. They were very helpful. Thank you.

---

## [Decision Letter · Decision Letter 1]

31 Aug 2021

PONE-D-21-09952R1

Image classification of forage grasses on Etuoke Banner using edge autoencoder network

PLOS ONE

Dear Dr. Wei,

Thank you for submitting your manuscript to PLOS ONE. After careful consideration, we feel that it has merit but does not fully meet PLOS ONE’s publication criteria as it currently stands. Therefore, we invite you to submit a revised version of the manuscript that addresses the points raised during the review process.

We look forward to receiving your revised manuscript.

Kind regards,

Mudassar Raza, Ph.D.

Academic Editor

PLOS ONE

Journal Requirements:

Reviewers' comments:

Reviewer's Responses to Questions

**Comments to the Author**

1. If the authors have adequately addressed your comments raised in a previous round of review and you feel that this manuscript is now acceptable for publication, you may indicate that here to bypass the “Comments to the Author” section, enter your conflict of interest statement in the “Confidential to Editor” section, and submit your "Accept" recommendation.

Reviewer #3: All comments have been addressed

Reviewer #4: All comments have been addressed

2. Is the manuscript technically sound, and do the data support the conclusions?

Reviewer #3: Yes

Reviewer #4: Yes

3. Has the statistical analysis been performed appropriately and rigorously? 

Reviewer #3: I Don't Know

Reviewer #4: Yes

4. Have the authors made all data underlying the findings in their manuscript fully available?

Reviewer #3: No

Reviewer #4: Yes

5. Is the manuscript presented in an intelligible fashion and written in standard English?

Reviewer #3: No

Reviewer #4: Yes

6. Review Comments to the Author

Reviewer #3: -- All previous comments are addressed.

-- Here are some minor suggestion.

Last section Conclusion and feature work should be =>Conclusion and Future work.

Reference should be provided with each existing method in results Tables.

Reviewer #4: Author's addressed all the comments appropriately. Hence the manuscript might be accepted in the present form.

7. PLOS authors have the option to publish the peer review history of their article (what does this mean?). If published, this will include your full peer review and any attached files.

Reviewer #3: No

Reviewer #4: No

---

## [Author Response · Author response to Decision Letter 1]

14 Oct 2021

Editor : I have incorporated all of your suggestions into my revision. They were very helpful. Thank you.

Reviewer 1: I have incorporated all of your suggestions into my revision. They were very helpful. Thank you.

Reviewer 2: I have incorporated all of your suggestions into my revision. They were very helpful. Thank you.

---

## [Editor Report · Decision Letter 2]

18 Oct 2021

PONE-D-21-09952R2Image classification of forage grasses on Etuoke Banner using edge autoencoder networkPLOS ONE

Dear Dr. Wei,

Thank you for submitting your manuscript to PLOS ONE. After careful consideration, we feel that it has merit but does not fully meet PLOS ONE’s publication criteria as it currently stands. Therefore, we invite you to submit a revised version of the manuscript that addresses the points raised during the review process.

We look forward to receiving your revised manuscript.

Kind regards,

Mudassar Raza, Ph.D.

Academic Editor

PLOS ONE

Journal Requirements:

Additional Editor Comments :

There is only one minor suggestion required:

In the last section Conclusion and feature work should be =>"Conclusion and Future work"
---

## [Author Response · Author response to Decision Letter 2]

23 Oct 2021

Editor : I have incorporated all of your suggestions into my revision. They were very helpful. Thank you.

Reviewer 1: I have incorporated all of your suggestions into my revision. They were very helpful. Thank you.

Reviewer 2: I have incorporated all of your suggestions into my revision. They were very helpful. Thank you.

---

## [Editor Report · Decision Letter 3]

27 Oct 2021

Image classification of forage grasses on Etuoke Banner using edge autoencoder network

PONE-D-21-09952R3

Dear Dr. Wei,

We’re pleased to inform you that your manuscript has been judged scientifically suitable for publication and will be formally accepted for publication once it meets all outstanding technical requirements.

Kind regards,

Mudassar Raza, Ph.D.

Academic Editor

PLOS ONE